# Trace Amine-Associated Receptor 1 Contributes to Diverse Functional Actions of O-Phenyl-Iodotyramine in Mice but Not to the Effects of Monoamine-Based Antidepressants

**DOI:** 10.3390/ijms22168907

**Published:** 2021-08-18

**Authors:** Ioannis Mantas, Mark J. Millan, Benjamin Di Cara, Lucianne Groenink, Sylvie Veiga, Laetitia Cistarelli, Mauricette Brocco, Marc Bertrand, Per Svenningsson, Xiaoqun Zhang

**Affiliations:** 1Department of Clinical Neuroscience, Karolinska Institute, 17177 Stockholm, Sweden; ioannis.mantas@ki.se (I.M.); xiaoqun.zhang@ki.se (X.Z.); 2Neuroscience and Inflammation Therapeutic Area, Institut de Recherches SERVIER, 78290 Croissy-sur-Seine, France; mark.john.millan@gmail.com (M.J.M.); benjamin.di-cara@servier.com (B.D.C.); sylvie.veiga@servier.com (S.V.); laetitia.cistarelli@servier.com (L.C.); mauri7b@outlook.fr (M.B.); 3Institute of Neuroscience and Psychology, College of Medicine, Vet and Life Sciences, Glasgow University, Glasgow G12 8QQ, UK; 4Department of Pharmacology, Utrecht Institute for Pharmaceutical Sciences, 3584 CS Utrecht, The Netherlands; l.groenink@uu.nl; 5Technologie Servier, 45000 Orléans, France; m.bertrand@servier.com

**Keywords:** TAAR1, o-PIT, DA, antidepressants

## Abstract

Trace Amine-Associated Receptor 1 (TAAR1) is a potential target for the treatment of depression and other CNS disorders. However, the precise functional roles of TAAR1 to the actions of clinically used antidepressants remains unclear. Herein, we addressed these issues employing the TAAR1 agonist, o-phenyl-iodotyramine (o-PIT), together with TAAR1-knockout (KO) mice. Irrespective of genotype, systemic administration of o-PIT led to a similar increase in mouse brain concentrations. Consistent with the observation of a high density of TAAR1 in the medial preoptic area, o-PIT-induced hypothermia was significantly reduced in TAAR1-KO mice. Furthermore, the inhibition of a prepulse inhibition response by o-PIT, as well as its induction of striatal tyrosine hydroxylase phosphorylation and elevation of extracellular DA in prefrontal cortex, were all reduced in TAAR1-KO compared to wildtype mice. O-PIT was active in both forced-swim and marble-burying tests, and its effects were significantly blunted in TAAR1-KO mice. Conversely, the actions on behaviour and prefrontal cortex dialysis of a broad suite of clinically used antidepressants were unaffected in TAAR1-KO mice. In conclusion, o-PIT is a useful tool for exploring the hypothermic and other functional antidepressant roles of TAAR1. By contrast, clinically used antidepressants do not require TAAR1 for expression of their antidepressant properties.

## 1. Introduction

Trace amines (TAs) are a diverse group of aminergic compounds that, in mammals, are either derived from the intake of food such as cheese, from the metabolism of thyroid hormone or from enzymatic pathways of synthesis in nervous tissue [1,2]. Enzymatic decarboxylation of the aromatic amino acids phenylalanine, tyrosine and tryptophan [3] respectively yields phenethylamine (PEA), tyramine and tryptamine [3]. In contrast to classical monoamines, such as noradrenaline (NA) and serotonin (5-HT), these and other TAs are present in far lower concentrations in brain and peripheral tissues [1,3]. TAs nonetheless appear to be functionally important in the control of mood, energetic status, cognition and other functions. Furthermore, it has been hypothesized that a disruption of their activity may contribute to the pathophysiology of multiple classes of metabolic, neuropsychiatric and even neurodegenerative disorders, in particular major depression and schizophrenia [1,3,4,5,6].

TAs have historically been considered “false transmitters”, since they displace monoamines from intraneuronal vesicles and encourage their reverse transport, actions thought to underlie sympathomimetic properties [7]. However, the notion that trace amines exert neuromodulatory effects, themselves, via specific classes of receptors [8] was concretized with the discovery of receptors that bind TAs in rodents [9,10], followed by the identification of a whole family of G protein-coupled “Trace Amine-Associated Receptors” (TAAR), now referred to as “TA” receptors [11]. Most classes of TAAR are highly expressed in the olfactory epithelium and act as volatile TA detectors [12,13]. Conversely, TAAR1 is mainly found in organs of the alimentary system and the brain [14]. In particular, TAAR1 is expressed in monoaminergic cell groups such as cortical and striatal projecting dopamine (DA) neurons, as well as corticolimbic projecting 5-HT neurons. Moreover, TAAR1 is found in the prefrontal cortex, entorhinal cortex, bed nucleus of stria terminalis, amygdala, hypothalamus and lateral parabrachial nucleus [14,15,16]. TAAR1 is coupled to Gs and G13 and responds to conventional, endogenous trace amines and amphetamine, methamphetamine and MDMA [10,17,18,19]. Due to its distinctive relationship with DA releasing agents, much TAAR1-related research has focused on the DAergic system. Mice lacking TAAR1 display higher VTA neuron firing rates [15,20,21] and heightened behavioural sensitivity to psychostimulants [15,16,18,22], suggesting that pharmacological TAAR1 activation might be beneficial for tackling schizophrenia. Indeed, TAAR1 agonists exhibit antipsychotic-like activity in certain animal models [23,24]. There is also evidence for deficits in trace amine activity in depression, and it has been reported that TAAR1 agonists improve affect, displaying antidepressant properties in rodents and nonhuman primates [23]. 

In view of their therapeutic promise, extensive efforts have been made to develop novel ligands for TAAR1 sites [25]. For example, the dual TAAR1/5-HT1A receptor agonist SEP-363856 showed antipsychotic properties in schizophrenic patients [26]. Even though larger trials are needed, SEP-363856 could become the first non-DA D2 receptor antagonist in clinical usage to exert antipsychotic efficacy [26]. Moreover, SEP-363856 displays negligible antipsychotic-related side effects such as weight gain, prolactin increase and occurrence of extrapyramidal symptoms [27]. In parallel with other TAAR1 agonists, preclinical studies have shown that SEP-363856 displays antidepressant actions in rodents [24], and beneficial effects of SEP-363856 in depressed patients may be revealed by future clinical trials. Consistent with this possibility, several studies have shown that trace amines exert a positive influence on affect, and that a reduction in their levels may be related to depressed states in patients [16,28,29]. Furthermore, it has been conjectured that a component of the antidepressant actions of Monoamine Oxidase inhibitors reflects their protection of trace amines from degradation [16,28,30]. That is, their antidepressant properties might at least partially depend upon increases in levels of trace amines and ipso facto activation of TAAR1 receptors—or another subclass of TAAR [31]. Alternatively, TAAR1 may act as a negative feedback regulator of trace amine levels, and TAAR1 antagonists might facilitate the antidepressant actions of monoamine oxidase inhibitor [16]. This remains, however, to be directly examined, and the putative role of TAAR1 in the actions of other classes of antidepressant agent has not as yet been evaluated [14,25].

Apart from conventional TAs, there is another TA group called thyronamines that display TAAR1 agonist properties. Unlike classical TAs, thyronamines are generated by enzymatic decarboxylation of thyroid hormones with ornithine decarboxylase [32,33]. In contrast to thyroid hormones, thyronamines reduce core temperature and cardiac output [34,35]. Further, TAAR1 was suggested to mediate these actions [35], and there is growing evidence for a close thyronamine-TAAR1 relationship [35,36,37]. Supporting this hypothesis, it was shown that 3-iodothyronamine (T1AM) increases striatal DA synthesis through TAAR1 signalling [38]. However, further research is needed to elucidate the precise role of TAAR1 in the actions of thyronamines. One synthetic derivative of the thyronamine family with high potency and selectivity for TAAR1 is o-phenyl-iodo-tyramine (o-PIT) [39]. Hence, o-PIT may be a useful tool for investigation of the TAAR1-mediated actions of thyronamines.

In view of the above observations, we exploited a neuroanatomical, neurochemical and behavioural approach to characterize the actions of o-Pit in TAAR1-KO mice. We particularly focused on readouts relevant to the influence of TAAR1 on DAergic neurotransmission and their potential role in the response to antidepressants. The data reveal a significant contribution of TAAR1 to several functional properties of o-PIT, including its actions in two “classic” antidepressant-responsive models, forced swimming and marble burying in mice. We also exploited these procedures to examine in complementary experiments whether TAAR1 sites are required for expression of the antidepressant actions of clinically employed monoaminergic agents.

## 2. Results

### 2.1. Exposure to O-PIT and Histoenzymological Detection of β-Galactosidase in TAAR1-KO Mice

The TAAR1-KO mice used in the present study, lack the coding sequence of TAA1 which is replaced by the LacZ reporter gene (Figure 1A). Brain and plasma levels of (exogenous) o-PIT were measured 60 min after intraperitoneal administration (Figure 1B–C). Considerable amounts of o-PIT were detected in the brain in a dose-dependent manner (2.5–40 mg/kg). Brain levels of o-PIT did not significantly differ between WT and TAAR1-KO mice (Figure 1B). ANOVA indicated a nonsignificant gen × drug interaction (F_2, 18_ = 0.7, *NS*) but a significant effect of drug (F_2, 18_ = 144.4, *p* < 0.01). Plasma levels of o-PIT were likewise not influenced by the genotype, and a nonsignificant gen × drug interaction (F_2, 17_ = 2.8, *NS*) but a significant effect of drug (F_2, 17_ = 55.4, *p* < 0.01) were also found (Figure 1C). In TAAR1-KO mice, X-gal deposits were detected in medial preoptic nucleus (MPO), medial division of bed nucleus of stria terminalis (BSTM), zona incerta and lateral hypothalamic area (ZI/LHA), basomedial and medial amygdaloid nucleus (BMA/MeA), ventral tegmental area (VTA) and substantia nigra pars compacta (SNC), layer 2 of lateral entorhinal cortex (LEnt), periaqueductal grey (PAG) and deep layers of superior colliculus (DpSC) and dorsal part of lateral parabrachial nucleus (LPBD) (Figure 1D). Our study did not include regions that lay caudal to pons.

### 2.2. Influence of O-PIT on Core Temperature in TAAR1-KO

It has been described that systemic administration of thyronamines reduces core body temperature in rodents (Figure 2A) [40]. Basal core temperatures were 37.9 ± 0.1 °C and 38.0 ± 0.1 °C in WT and TAAR1-KO mice, respectively, and were not statistically different between genotypes. Administration of o-PIT dose-dependently decreased core temperature in WT and TAAR1-KO mice, but its action was less marked in TAAR1-KO mice (maximal fall to 34.8 ± 0.2 °C vs. 33.2 ± 0.6 °C in WT mice) (Figure 2B). ANOVA indicated a significant gen × drug interaction (F_3, 54_ = 2.8, *p* < 0.05), effect of drug (F_3, 54_ = 78.0, *p* < 0.01) and effect of gen (F_1, 54_ = 12.3, *p* < 0.01). Moreover, the influence of different classes of monoaminergic antagonists upon the hypothermia induced by o-PIT (20 mg/kg, i.p.) was evaluated (Figure 2C). Pretreatment with the α2-adrenoceptors antagonist, RX821,002, which was inactive by itself (gen × pret interaction: F_5, 89_ = 0.6, *NS*), dose-dependently attenuated the o-PIT-induced hypothermia both in WT and TAAR1-KO mice. The ability of RX821,002 to reverse the effect of o-PIT was comparable between genotype as signed by a nonsignificant gen × pret × drug interaction (F_4, 154_ = 1.1, *NS*).

### 2.3. Influence of O-PIT on Prepulse Inhibition in TAAR1-KO Mice

O-PIT significantly increased the percent of prepulse inhibition (Figure 3A) in WT mice but was inactive in TAAR1-KO mice, as indicated by a significant gen × drug interaction (F_1, 44_ = 5.1, *p* < 0.05; Figure 3B). The effect of o-PIT was independent of prepulse intensity (gen × drug × intensity interaction (F_3, 132_ = 1.0, *NS*)). O-PIT had no effect on the basal startle response in either genotype (drug effect: F_1, 44_ = 1.2, *NS*; gen × drug interaction: F_1, 44_ = 1.0, *NS*; Figure 3C).

### 2.4. Induction by O-PIT of Tyrosine Hydroxylase Phosphorylation (TH) and Activity in the Caudoputamen (CPu)

To investigate the role of o-PIT–TAAR1 interaction in DAergic system, o-PIT was applied in CPu coronal slices, which were subsequently used for WB analysis (Figure 4A). O-PIT concentration dependently enhanced the Ser^19^ phosphorylation of TH in the CPu of WT but not TAAR1-KO mice (gen effect: F_3, 40_ = 3.052, *p* < 0.05; Figure 4B–C). Similar trends, although not significant, were found on Ser^31^ and Ser^40^ in WT and TAAR1-KO mice (Ser^31^
gen effect: F_3, 39_ = 3.39, *NS*; Ser^40^
gen effect: F_3, 40_ = 1.25, *NS*; Figure 4D–E). In all cases, the effects were biphasic, with a further increase in concentrations leading to an inversion of the concentration–response curve.

### 2.5. Influence of O-PIT upon Dialysis Levels of DA, NA and 5-HT in the mPFC of Freely Moving TAAR1-KO Mice

A microdialysis probe was implanted in mPFC, which contains DA, 5-HT, noradrenaline (NA) and acetylcholine afferents (Figure 5A). No change in basal extracellular levels of DA, 5-HT or NA was detected in the frontal cortex of TAAR1-KO mice (Table 1). Injection of 10 mg/kg o-PIT elicited a sustained elevation in DA levels in WT but not TAAR1-KO mice, as indicated by a significant time × gen × drug interaction (F_18, 168_ = 2.6, *p* < 0.01; Figure 5B). Subsequent analyses indicated that o-PIT was active in WT mice (time × drug interaction: F_8, 96_ = 3.5, *p* < 0.01) but was inactive in TAAR1-KO mice (F_8, 72_ = 1.9, *NS*). However, at the dose of 20 mg/kg, o-PIT elicited a significant increase in DA levels in TAAR1-KO mice (AUC; Figure 5C), suggesting that the effect of o-PIT was blunted in TAAR1-KO mice (gen × drug interaction: F_3, 40_ = 2.67, *p* < 0.05). Underlining a specific action of o-PIT upon DAergic transmission, levels of 5-HT and NA, detected concomitantly, were not altered by 10 mg/kg o-PIT (Figure 5D–E). Likewise, o-PIT was without effect upon acetylcholine and amino acids levels (Table 1). Finally, evaluated upon subcortical monoaminergic transmission, the effect of 10 mg/kg o-PIT was comparable between WT and TAAR1-KO mice (Table 1).

### 2.6. Influence of Classical Antidepressants upon Dialysis Levels of DA, NA and 5-HT in the mPFC of Freely Moving TAAR1-KO Mice

WT and TAAR1-KO mice were injected with 10 mg/kg of Venlafaxine, Citalopram, Bupropion, Selegiline or Reboxetine, and levels of monoamines were measured by microdialysis in mPFC (Figure 6A). DA, NA and 5-HT were significantly affected by antidepressant drug treatment (drug effect: F_5, 63_ = 34.9, *p* < 0.01; F_5, 66_ = 34.96, *p* < 0.01; F_5, 65_ = 27.15, *p* < 0.01, respectively; Figure 6B–D). However, DA, NA and 5-HT did not display any significant genotype effect (gen effect: F_1, 63_ = 2.8, *NS*; F_1, 66_ = 0.99, *NS*; F_1, 65_ = 0.52, *NS*, respectively; Figure 6B–D). Post hoc analysis revealed that Venlafaxine and Citalopram increased 5-HT levels, Selegiline increased DA and NA levels and Reboxetine increased NA and 5-HT levels (Figure 6B–D). These effects were observed in both WT and TAAR1-KO mice (Figure 6B–D).

### 2.7. Influence of O-PIT and Commonly Used Antidepressants on the Forced-Swim Test in TAAR1-KO Mice

In the forced-swim test (Figure 7A), TAAR1-KO mice showed a spontaneous but not reproducible decrease in total time of immobility, which was 210.0 ± 10.0 sec in WT mice and 154.6 ± 21.7 sec in TAAR1-KO mice (F_1, 15_ = 6.7, *p* < 0.01; Figure 7B). The effect of o-PIT was influenced by the genotype, as indicated by the significant gen × drug interaction (F_3, 51_ = 2.9, *p* < 0.05). The time of immobility was dose-dependently (2.5–40 mg/kg) reduced by o-PIT in WT mice (F_3, 29_ = 5.8, *p* < 0.01) but was absent in TAAR1-KO mice (F_3, 22_ = 0.5, NS). In contrast, TAAR1-KO mice displayed similar reduction of immobility time compared to the WT mice after the administration of duloxetine, venlafaxine, maprotiline, fluoxetine, bupropion, radafaxine, selegiline and reboxetine (gen × drug interaction: F_3, 40_ = 1.2, *NS*; F_3, 79_ = 0.3, *NS*; F_4, 54_ = 0.8, *NS*; F_3, 46_ = 1.4, *NS*; F_4, 68_ = 0.8, *NS*; F_4,62_ = 0.7, *NS*; F_3,46_ = 0.05, *NS* and F_6,91_ = 0.05, *NS*, respectively; Figure 7C–J).

### 2.8. Influence of O-PIT and Commonly Used Antidepressants on Marble-Burying Test in TAAR1-KO Mice

In the marble-burying test, TAAR1-KO mice displayed a constitutive alteration in the performance, with a total of 15.3 ± 1.2 buried marbles in WT mice and 18.4 ± 0.6 in TAAR1-KO mice (F_1, 29_ = 6.0, *p* < 0.05; Figure 8A). A dose-dependent (2.5–40 mg/kg) suppression of the number of buried marbles was provoked by o-PIT, but its effect was considerably attenuated in TAAR1-KO mice. ANOVA indicated a significant gen × drug interaction (F_4, 73_ = 6.2, *p* < 0.01), effect of gen (F_1, 73_ = 40.1, *p* < 0.01) and effect of drug (F_4, 73_ = 79.3, *p* < 0.01). Even though there was no significant gen × drug interaction for clomipramine (F_4, 51_ = 0.9, *NS*; Figure 8C) and amitriptyline (F_3, 66_ = 2.3, *NS*; Figure 8D), TAAR1-KO mice exhibit a slight rightward shift in the dose-response curve of tricyclic antidepressants, which is similar to that one observed after o-PIT administration. Particularly, there was a significant effect in both gen and drug after the administration of clomipramine (gen effect: F_1, 51_ = 19.22, *p* < 0.01; drug effect: F_4, 51_ = 20.34, *p* < 0.01) and amitriptyline (gen effect: F_1, 66_ = 17.86, *p* < 0.01; drug effect: F_3, 66_ = 48.08, *p* < 0.01). TAAR1-KO mice displayed similar reduction of the number of buried marbles to the WT mice after the administration of duloxetine, venlafaxine, fluoxetine, citalopram, bupropion, radafaxine, selegiline and reboxetine (gen × drug interaction: F_3.44_ = 1.5, *NS*; F_3, 54_ = 0.8, *NS*; F_3, 43_ = 0.5, *NS*; F_3, 41_ = 2.6, *NS*; F_4, 47_ = 0.7, *NS*; F_4, 54_ = 0.3, *NS*; F_3, 47_ = 1.5, *NS* and F_3, 51_ = 0.6, *NS*, respectively; Figure 8E–L).

### 2.9. Influence of O-PIT and Monoaminergic Receptor Antagonists on Marble-Burying Test in TAAR1-KO Mice

High doses of o-PIT (>20 mg/kg) display robust anticompulsive effects in both WT and TAAR1-KO mice. This indicates that, apart from TAAR1, o-PIT may act on additional receptor targets. D2 (Raclopride), α2 (RX821002), 5-HT1A (WAY100635), 5-HT2A (MDL100907) and 5-HT2C (SB206553) antagonists were co-administered together with o-PIT (Figure 9A). Even though there was a significant drug effect for all the performed experiments (drug effect: F_3, 38_ = 54.96, *p* < 0.01; F_3, 38_ = 54.96, *p* < 0.01; F_3, 47_ = 16.15, *p* < 0.01; F_3, 45_ = 34.62, *p* < 0.01; F_3, 41_ = 55.25, *p* < 0.01, respectively; Figure 9B–F), post hoc test revealed that all the antagonists that were used failed to abolish the anticompulsive effects of o-PIT in both WT and TAAR1-KO animals.

## 3. Discussion

### 3.1. Hypothermia Elicited by O-PIT Is Partially Mediated by TAAR1

One major effect of thyronamines is to counteract the hyperthermic effects of thyroid hormones [35]. Consistent with an action in the brain, intracerebroventricular as well as systemic injection of T1AM induces hypothermia [41]. The major brain centre that actively modulates core temperature is the MPO [42]. This hypothalamic brain area harbours warm-sensitive neurons, which act as the body’s thermostat [43]. Accordingly, direct T1AM application in MPO decreases body temperature, indicating that the cooling effect of thyronamines may stem from their activity on warm-sensitive neurons [44]. Supporting this possibility, we demonstrate here a high density of TAAR1 expressing cells in MPO. However, a previous study has reported that T1AM reduces body temperature in TAAR1-KO mice [45]. Moreover, like thyronamines, direct application of clonidine, an α2 agonist, in MPO causes hypothermia [46]. TAAR1 and α2-adrenoceptors share similar pharmacophore space [21], and T1AM is also a ligand for α2-adrenergic receptor [36]. Hence, it was suggested that T1AM-induced hypothermia is at least partially mediated by α2 expressing neurons in MPO [47]. Here, we report that o-PIT induces hypothermia and that this effect was blunted in TAAR1-KO mice supporting a role for TAAR1 in the mediation of hypothermia. Interestingly, in light of the above remarks, the selective α2-antagonist, RX821002, partially counteracted the hypothermic effect of o-PIT in both WT and TAAR1-KO mice, indicating an additional role for α2 adrenergic receptors in the effect of o-PIT on body temperature. Finally, since o-PIT caused some reduction of body temperature, even in RX821002-treated TAAR1-KO mice, additional, currently unknown targets may also mediate its hypothermic actions. Further work is warranted to further clarify the mechanisms underlying the influence of o-PIT and TAAR1/trace amines, more generally, upon core temperature, an important issue in view of their links to thyroid hormones [34,35].

### 3.2. O-PIT Enhancement of Prepulse Inhibition Is TAAR1-Dependent

O-PIT enhanced prepulse inhibition (PPI) in wildtype mice but not in TAAR1-KO mice. Stimulation of striatal D2 signalling disrupts sensorimotor gating [48,49,50], and schizophrenic patients display impairments in sensorimotor gating [51]. Hence, TAAR1’s ability to diminish the activity of VTA dopamine neurons projecting to NAc may underlie the action of o-PIT on PPI. These data provide further evidence that TAAR1 agonism may be a novel antipsychotic approach. Additionally, the current study reveals a dense positive population of TAAR1 in LEnt, which is crucial for performance of sensorimotor gating tasks [52,53]. It has been described that T1AM–TAAR1 signalling in entorhinal slices prevents ischemia-induced synaptic depression [54]. Thus, o-PIT may act on TAAR1 expressing cells in LEnt to increase prepulse inhibition.

### 3.3. O-PIT Increases TH Phosphorylation through TAAR1 in the Striatum

O-PIT significantly stimulated Ser^19^-TH phosphorylation in striatum. There was also a trend for o-PIT to increase Ser^40^-TH. Ser^19^ phosphorylation of TH displays faster kinetics than Ser^40^, which may explain the observed effects [55]. The stimulatory effects of o-PIT on TH phosphorylation were abolished in TAAR1-KO mice. In agreement with these data, T1AM stimulates TH phosphorylation through TAAR1-dependent Gs signalling [38]. In this context, it is interesting to note that the endogenous TAAR1 agonist, tyramine, actually inhibits TH phosphorylation in a TAAR1-independent manner [38]. It is established that tyramine displays amphetamine-like properties and causes strong DA release [1,56,57,58], which probably masks TAAR1-dependent actions on TH phosphorylation [38]. However, thyronamines are TAAR1 agonists devoid of amphetamine-like properties and appear suitable for studying TAAR1-dependent effects on DA system.

### 3.4. O-PIT Stimulates DA Release in mPFC through TAAR1

Dysfunction of PFC is strongly associated with mental illnesses such as MDD [59]. An increase in frontocortical DA transmission would be expected to have a beneficial impact upon both the affective and cognitive deficits in MDD [60,61]. Since TH is relatively lowly expressed in PFC, we could not measure TH phosphorylation in this region. However, using in vivo microdialysis in freely moving animals, we found that o-PIT increases DA release in mPFC in a manner that is dependent on TAAR1. Since TAAR1 is expressed in raphe nuclei, it could be anticipated that o-PIT may promote 5-HT release and phosphorylation of tryptophan hydroxylase 2. However, the present study shows that o-PIT effects upon monoamine release in mPFC are limited to the DAergic system.

### 3.5. A Differential Role of TAAR1 in the Antidepressant Actions of O-PIT vs. Clinically used Classes of Antidepressants

Behavioural tests that are used to evaluate antidepressantdrug efficacy include forced-swim and marble-burying tests. Each test does not examine depression or compulsion, per se, but rather pragmatic behavioural responses to clinically used agents for these disorders. There is evidence that TAAR1 agonists exhibit antidepressant actions in animal depression-like behavioural tests [23]. Likewise, the current study shows that o-PIT exhibits antidepressant action, which is diminished, at least at lower doses, in TAAR1-KO mice. A detailed pharmacological study was performed to better understand the action of o-PIT in the marble-burying test. However, the effects of o-PIT in this test are not blocked by antagonism of D2, α2, 5-HT1A, 5-HT2A or 5-HT2C receptors. Similar to o-PIT-induced hypothermia, the exact class of receptor that are responsible for the effects of higher o-PIT doses in this test requires further investigation. In contrast, the present data demonstrates that the efficacy of commonly used monoamine-based antidepressants is not altered in TAAR1-KO mice. Recently, we reported that the irreversible and nonselective MAO A/B inhibitor, tranylcypromine, display increased antidepressant efficacy in TAAR1-KO mice [16]. This may relate to tyramine build-up and general stimulation not seen with most antidepressants [16]. To fully understand the role of TAAR1 in mediating antidepressant actions, additional behavioural tests will be important to perform in the future. Nevertheless, corroborating the behavioural data, there was no influence of TAAR1 on the stimulatory effects of the clinically used antidepressants on extracellular monoamine levels in mPFC measured by microdialysis in freely moving animals.

## 4. Conclusions

This study demonstrates that o-PIT regulates, via TAAR1-dependent mechanisms, core temperature, sensorimotor gating, striatal TH phosphorylation, mPFC DA release and antidepressant behavioural responses. In contrast, the behavioural and neurochemical actions of several classes of clinically used antidepressants do not involve TAAR1. These data suggest that o-PIT is a useful experimental tool for studies of TAAR1-dependent effects, particularly related to the DA system, and that o-PIT exerts antidepressant actions in a manner distinct from monoamine-based antidepressant. It will be interesting in future work to examine antidepressant actions of o-PIT in models that are resistant to monoamine-based antidepressants.

## 5. Material and Methods

### 5.1. Animals

Homozygous TAAR1-KO mice lack the single intron-free Taar1 coding exon, which is replaced with an IRES-LacZ reporter gene [18]. TAAR1-KO mice were backcrossed on a pure C57BL/6J genetic background for 8 generations (Charles River). In all experiments, 5–8 weeks old male mice were used, and the experimenters were blinded as regard to the genotype and treatment. They were housed 4–5 per cage under a 12 : 12 h light/dark cycle and had free access to food and water. Housing and experimental procedures were fully compliant with the principles of Care and Use of Laboratory Animals. All experiments were conducted in accordance with European/EU (Directive 2010/63) directives on animal experimentation. 

### 5.2. Drugs

*o*-phenyl-3-iodotyramine hydrochloride (o-PIT) was synthesized by Servier chemists. Raclopride, RX821,002, WAY100,635, MDL100907 and SB206553 were purchased from Sigma. Compounds were dissolved in sterile water. MDL100907 and SB206553 were dissolved in sterile water plus lactic acid, and the pH was adjusted near neutrality. Sterile water with or without lactic acid were used as vehicle control (VEH). All solutions were injected in a volume of 10 mL/kg. Dose was expressed in terms of free base.

### 5.3. Plasma and Brain Levels of O-PIT

Blood samples (kept from coagulation with heparin) and whole brains (without cerebellum) were immediately taken sixty minutes after administration of o-PIT (2.5–20 mg/kg, i.p.). Plasma (obtained from blood at 800 rpm) and homogenized brains samples were assayed for o-PIT content using mass spectrometry. The detection limit was 5 ng/mL.

### 5.4. Histoenzymological Detection of β-Galactosidase

Frozen perfused (2% paraformaldehyde, 0.2% glutaraldehyde in PBS) brains were cryostat cut in 30 µm coronal sections. The sections were washed 2 times for 5 min with phosphate buffer solution (PBS). Then, they were incubated overnight in working buffer solution prepared according to manufacturer instructions (GALS, Sigma–Aldrich). The sections were washed 3 times for 5 min in PBS. They were mounted with aqueous media and then digitalized with NanoZoomer (Hamamatsu, France).

### 5.5. Core Temperature Measurement

Experiments started at 10.00 a.m. and were randomized. Core temperature was determined employing a rectal thermoprobe [62] and was measured in basal condition and 30, 60 and 120 min after o-PIT administration. In interaction studies, RX821,002 was administered 30 min before o-PIT (20 mg/kg; i.p.).

### 5.6. Acoustic Startle and Prepulse Inhibition

Startle reflexes were measured in mice as previously described [63]. Briefly, they were detected in nonrestrictive Plexiglas cylinder (SR-LAB; San Diego Instruments, CA, USA). Mice were familiarized with the startle procedure 1 week before testing. Startle stimuli (110 dB, 50 msec), alone, or preceded by noise prepulses (20 msec) of 2, 4, 8 or 16 dB above background (70 dB) were presented through a high-frequency speaker. Startle magnitudes were immediately sampled for 65 msec (1 kHz). A startle response was defined as the peak response of the 65 msec. Three consecutive blocks of test trials were as follows: blocks 1 and 3 consisted of 6 startle stimulus, alone trials; block 2 consisted of 12 startle stimulus, alone trials; 10 startle prepulse trials per prepulse intensity and 10 no-stimulus trials.

### 5.7. Tyrosine Hydroxylase Phosphorylation in Brain Slices

Striatal slices of mouse brain (300 μm, Leica Vibratome) were preincubated in Krebs buffer (118 mM NaCl, 4.7 mM KCl, 1.5 mM Mg_2_SO_4_, 1.2 mM KH_2_PO_4_, 25 mM NaHCO_3_, 11.7 mM glucose, 1.3 mM CaCl_2_) at 30 °C under constant oxygenation (95% O_2_/5% CO_2_) for 60 min, with a change of buffer after 30 min. They were then treated with MDMA for 5 min. The buffer was then removed, and the slices were rapidly frozen, sonicated in 1% SDS and boiled for 10 min. Amounts of protein in homogenates were determined using the bicinchoninic acid protein assay method (Pierce, Rockford, IL, USA). Equal amounts of protein were processed by using 10% acrylamide gels. Western blot (WB) was carried out with phosphorylation state-specific antibodies against P-Ser^19^-TH, P-Ser^31^-TH and P-Ser^40^-TH (Millipore, Solna, Sweden) or antibodies that are not phosphorylation state-specific against total TH and actin (Millipore). Antibody binding was detected by enhanced chemiluminescence (ECL; GE Healthcare, Uppsala, Sweden) and quantified by densitometry (NIH IMAGE 1.61 software). Data are percentage of total levels.

### 5.8. Microdialysis in Freely Moving Mice

The microdialysis probe was surgically implanted in medial prefrontal cortex (mPFC) as previously described [18]. The effect of o-PIT upon the different classes of neurotransmitters was evaluated in separate experiments. (1) Monoamines were quantified using an electrochemical procedure [18]. The limit of sensitivity was 0.2 pg. (2) Acetylcholine was quantified in the absence of acetylcholinesterase inhibition [64]. The limit of sensitivity was 0.5 pg. (3) Amino acids were subjected to a precolumn derivatization using naphthalene dicarboxaldehyde 5 mM and sodium cyanide (NaCN 10 mM, borate buffer 0.1 M, pH 9.5) for 5 min at 21 °C. Fluorescent products of the reaction (5 µL) were injected. The mobile phase was composed of ammonium acetate 50 mM, tetrahydrofuran–acetonitrile (gradient), pH 6.8, and delivered at 0.35 mL/min. Amino acids were separated on a C18 column (Hypersil, 250 × 2.1 mm, 5 µm) at 44 °C. A fluorimetric detector (Jasco FP2020+) settled at λ_Ex_ 420 nm and λ_Em_ 490 nm was used. The limit of sensitivity was 1.0 pg. The standard trapezoidal method was used for the calculation of AUC (% × min × 10^−3^, arbitrary units). In further experiments, the effects of diverse classes of clinically employed antidepressant agent were examined.

### 5.9. Forced-Swim Test

As previously described [65], mice were placed in individual glass cylinders (24 cm height; 12 cm diameter) containing 6 cm of water at 24 ± 0.5 °C for 6 min. Duration of immobility was measured over the last 4 min. o-PIT was administered (s.c.) 30 min prior testing.

### 5.10. Marble-Burying Behaviour

The procedure was described previously [65]. Mice were injected with o-PIT (s.c.) and placed 30 min later in transparent polycarbonate cages (45 × 30 × 19 cm) containing a 5 cm layer of sawdust and 24 glass marbles (1.5 cm diameter) evenly spaced against the wall of the cage. The number of marbles buried at least two-thirds in the sawdust was recorded 30 min later. In interaction studies, raclopride, RX821,002, WAY100,635, MDL100907 and SB206553 were administered 30 min before o-PIT (20 mg/kg; i.p.).

### 5.11. Statistical Analysis

Statistical comparisons of the effect of each drug in the two groups were achieved using two-way ANOVA with genotype (gen) and drug as between factors and was followed by the Tukey, Sidak or Bonferroni post hoc tests for multiple comparisons. The following three-way ANOVA were used: *plus* pretreatment (pret) as between factor in drug interaction studies; *plus*
time as within factor and repeated measurement in kinetic studies of the effect of o-PIT and *plus*
intensity as repeated within factor in PPI studies. Values of *p* < 0.05 were considered significant.

## Figures and Tables

**Figure 1 ijms-22-08907-f001:**
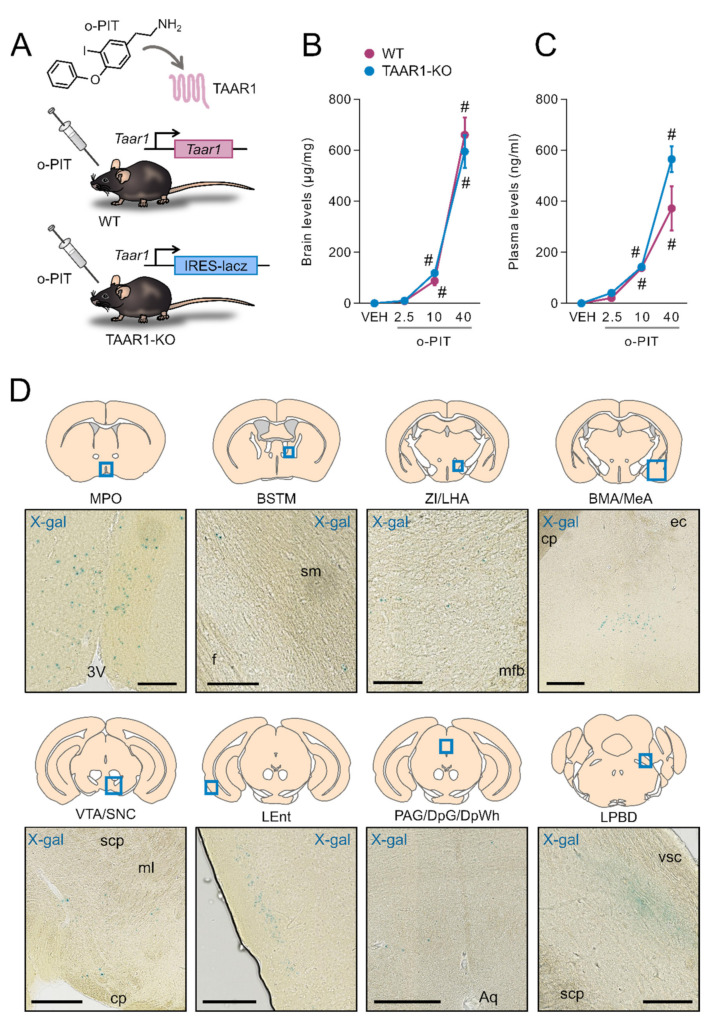
O-PIT brain and plasma pharmacokinetics in TAAR1-KO mice and TAAR1 distribution in the brain. (**A**) Chemical structure of o-PIT and schematic representation of the TAAR1-KO mouse genetic construct. (**B**–**C**) Levels of o-PIT were quantified in brain (**B**) and plasma (**C**) 60 min after administration. Levels are expressed as absolute values per weight of brain sample (µg/mg) and volume of plasma (ng/mL). All data are means ± SEM. N = 4–5 per group. *p* < 0.05, influence of treatment (#) or genotype (*). VEH: vehicle. (**D**) Neuroanatomical distribution of TAAR1 expressing cells using β-galactosidase detection in TAAR1-KO mice sections and that the staining represents expression under the endogenous TAAR1 promoter (N = 3). MPO: medial preoptic nucleus (scale bar: 100 μm), BSTM: medial bed nucleus of stria terminalis (scale bar: 100 μm), ZI/LHA: zona incerta/lateral hypothalamic area (scale bar: 100 μm), BM/Me: basomedial/medial amygdaloid nucleus (scale bar: 250 μm), VTA/SNC: ventral tegmental area/compact part of substantia nigra (scale bar: 250 μm), LEnt: lateral entorhinal cortex (scale bar: 250 μm), PAG/DpG/DpWh: periaqueductal grey/deep grey layer of superior colliculus/deep white layer of superior colliculus (scale bar: 250 μm), LPBD: dorsal part of lateral parabrachial nucleus (scale bar: 100 μm), 3V: 3rd ventricle, sm: stria medullaris, f: fornix, mfb: medial forebrain bundle, cp: cerebral peduncle, ec: external capsule, scp: superior cerebellar peduncle, ml: medial lemniscus, Aq: aqueduct.

**Figure 2 ijms-22-08907-f002:**
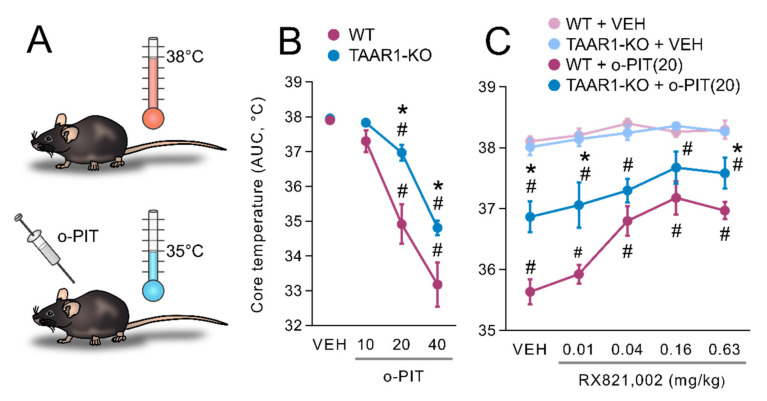
Drug interaction with o-PIT upon core temperature in TAAR1-KO and WT mice. (**A**) Schematic depiction of hypothermic effects of o-PIT. (**B**) Core temperature is expressed as area under the curve (AUC) calculated over the 120 min following administration of o-PIT (2.5–40 mg/kg). (**C**) Antagonists (mg/kg, s.c.) were administered 30 min prior to o-PIT (20 mg/kg). Core temperature was determined 60 min following o-PIT. (**C**) Dose-dependent (0.01–0.63 mg/kg) effect of RX821, 002 upon the effect of o-PIT. All data are means ± SEM. N = 6–8 per group. *p* < 0.05, influence of treatment (#) or genotype (*). VEH: vehicle.

**Figure 3 ijms-22-08907-f003:**
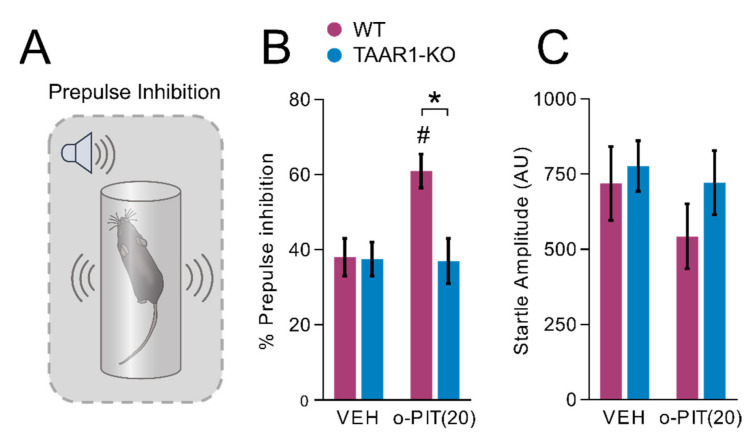
Influence of o-PIT on prepulse inhibition in TAAR1-KO and WT mice. (**A**) Schematic depiction of prepulse inhibition experiment. (**B**,**C**) Show that the enhancement by o-PIT of prepulse inhibition (**B**) is specific in that it does not affect the startle reflex, per se, neither in WT nor in TAAR1-KO mice (**C**). Data represent means ± SEM. N = 8–12 per group. *p* < 0.05, influence of treatment (#) or genotype (*). VEH: vehicle.

**Figure 4 ijms-22-08907-f004:**
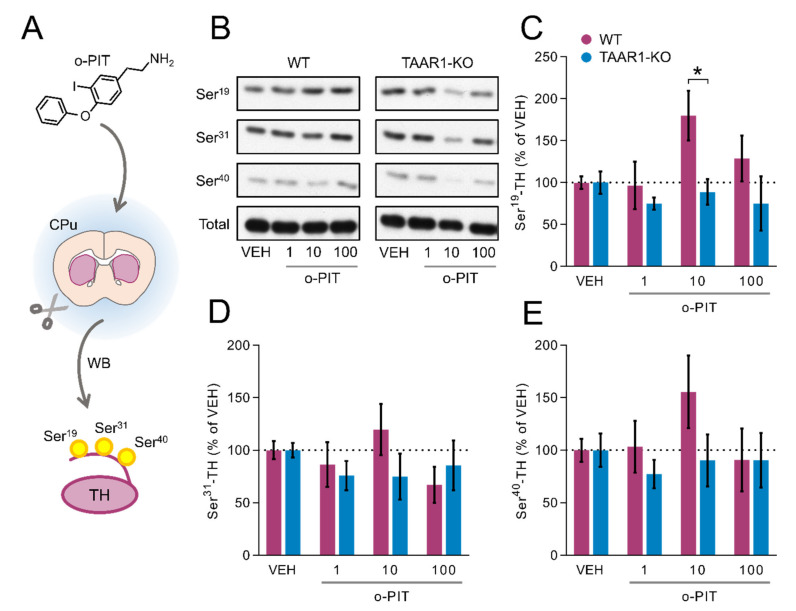
Influence of o-PIT on Tyrosine Hydroxylase (TH) phosphorylation *in CPu* of TAAR1-KO and WT mice. (**A**) Schematic depiction of experimental procedure. (**B**) Immunoblots of P-Ser^19^-TH, P-Ser^31^-TH, P-Ser^40^-TH and total-TH were quantified following increasing concentrations of o-PIT (1–100 µM). (**C**–**E**) Bar graphs showing the immunoblot quantification of P-Ser^19^-TH (**C**), P-Ser^31^-TH (**D**), P-Ser^40^-TH (**E**). CPu TH activity was measured following the perfusion of 10 µM o-PIT. Data are expressed as percentage of vehicle (100%). All data are means ± SEM. N = 4–5 per group. *p* < 0.05, influence of treatment (#) or genotype (*). VEH: vehicle.

**Figure 5 ijms-22-08907-f005:**
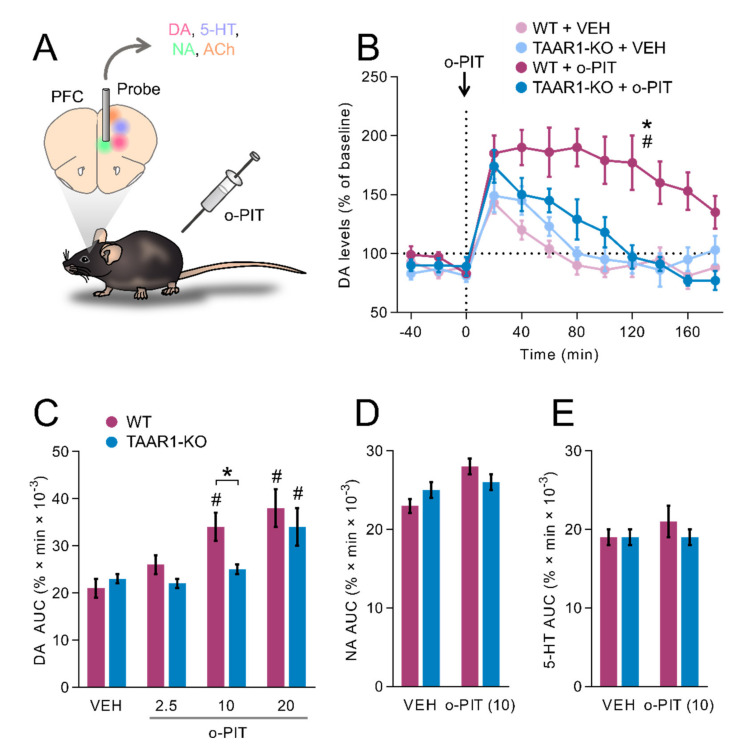
Effect of o-PIT (10 mg/kg) on extracellular levels of DA, 5-HT and NA in the mPFC of TAAR1-KO and WT mice. (**A**) Schematic depiction of experimental procedure. (**B**) The vertical dotted line indicates the time at which o-PIT (10 mg/kg) was administered. DA levels are expressed as a percentage of baseline (100%). (**C**–**E**) Histograms show values of area under the curve (AUC _[0..180]_, arbitrary units) for the dose–response of o-PIT upon DA (**C**) and for the effect of 10 mg/kg o-PIT on NA (**D**) and 5-HT (**E**). All data are means ± SEM. N = 6–8 per group. *p* < 0.05, influence of treatment (#) or genotype (*). VEH: vehicle.

**Figure 6 ijms-22-08907-f006:**
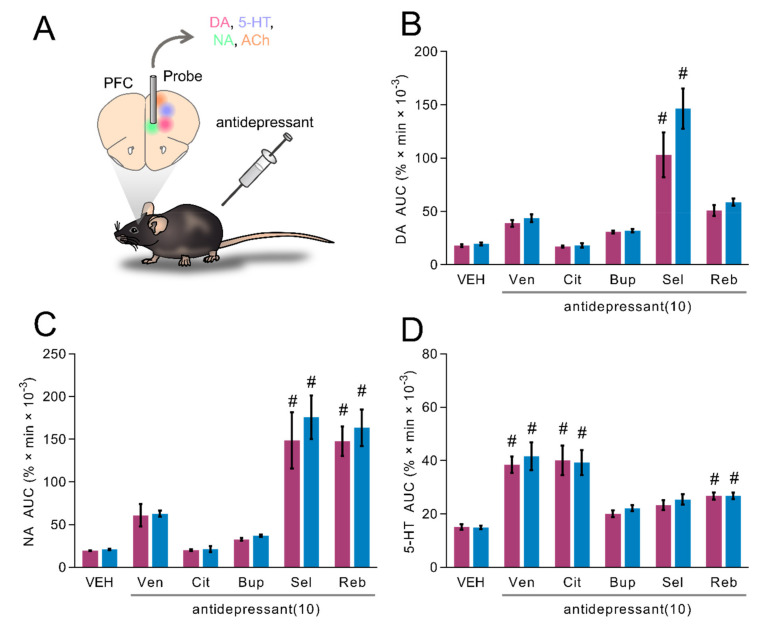
Effects of different antidepressants (10 mg/kg) on extracellular levels of DA, 5-HT and NA in the mPFC of TAAR1-KO and WT mice. (**A**) Schematic depiction of experimental procedure. (**B**–**D**) Histograms show values of areas under the curve (AUC _[0 ..180]_, arbitrary units) for the response of venlafaxine (Ven), citalopram (Cit), bupropion (Bup), selegiline (Sel) and reboxetine (Reb) upon DA (**A**), NA (**B**) and 5-HT (**C**). All data are means ± SEM. N = 6–8 per group. *p* < 0.05, vehicle + antidepressant (#). VEH: vehicle.

**Figure 7 ijms-22-08907-f007:**
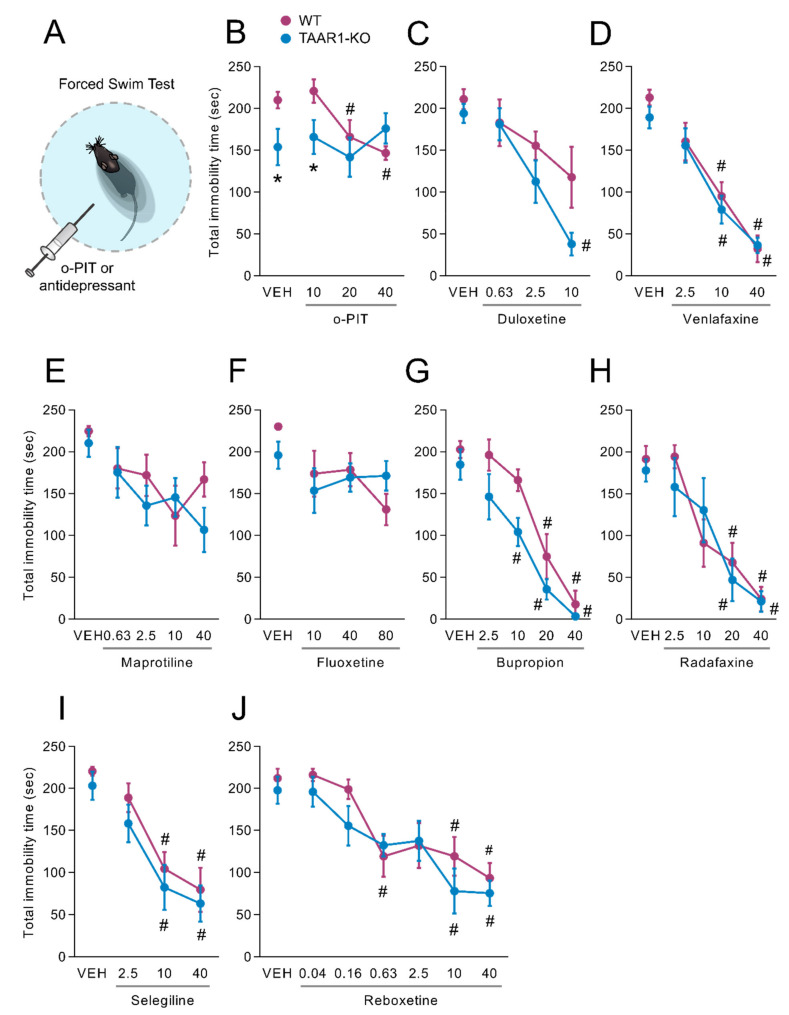
Influence of o-PIT on forced-swim test in TAAR1-KO and WT mice. (**A**) Schematic depiction of forced-swim test. (**B**–**J**) Values correspond to the duration of immobility during 4 min test after acute o-PIT treatment (2.5–40 mg/kg) (**B**), duloxetine (0.63–10 mg/kg) (**C**), venlafaxine (2.5–40 mg/kg) (**D**), maprotiline (0.63–40 mg/kg) (**E**), fluoxetine (10–80 mg/kg) (**F**), bupropion (2.5–40 mg/kg) (**G**), radafaxine (2.5–40 mg/kg) (**H**), selegiline (2.5–40 mg/kg) (**I**) and reboxetine (0.04–40 mg/kg) (**J**). Data represent means ± SEM. N = 8–12 per group. *p* < 0.05, influence of treatment (#) or genotype (*). VEH: vehicle.

**Figure 8 ijms-22-08907-f008:**
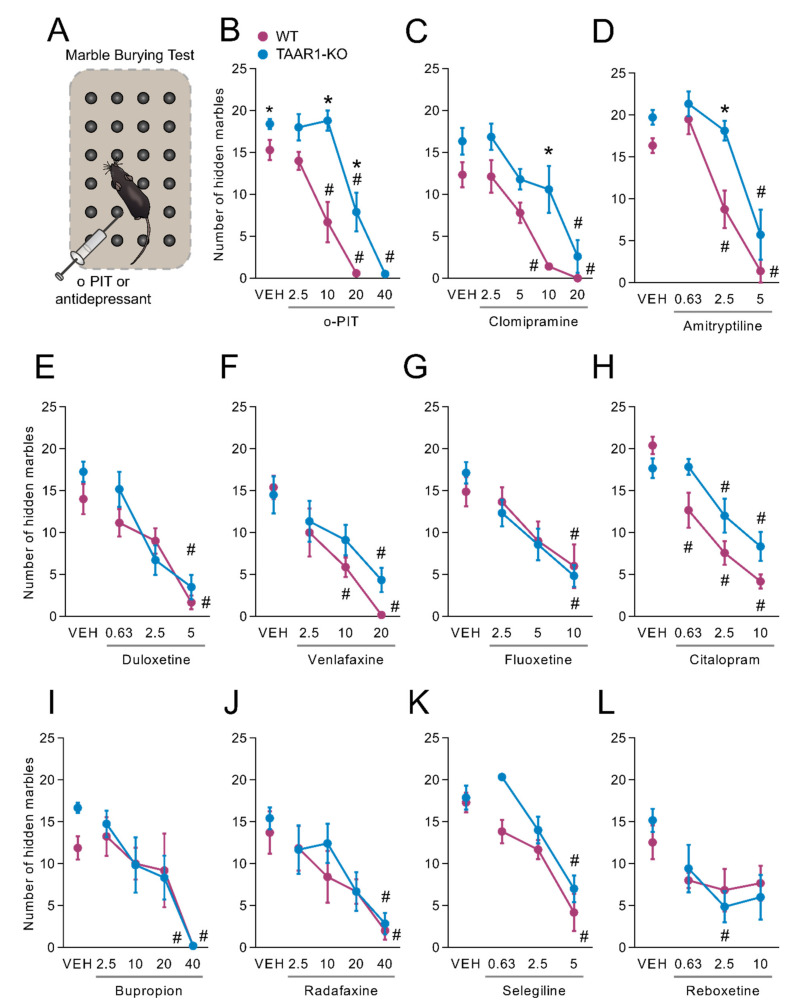
Influence of o-PIT on marble-burying in TAAR1-KO and WT mice. (**A**) Schematic depiction of marble-burying test. (**B**–**L**) Values correspond to the number of hidden marbles over 30 min after acute o-PIT treatment (2.5–40 mg/kg) (**B**), clomipramine (2.5–20 mg/kg) (**C**), amitriptyline (0.63–5 mg/kg) (**D**), duloxetine (0.63–5 mg/kg) (**E**), venlafaxine (2.5–20 mg/kg) (**F**), fluoxetine (2.5–10 mg/kg) (**G**), citalopram (0.63–10 mg7 kg) (**H**), bupropion (2.5–40 mg/kg) (**I**), radafaxine (2.5–40 mg/kg) (**J**), selegiline (0.63–5 mg/kg) (**K**) and reboxetine (0.63–40 mg/kg) (**L**). Data represent means ± SEM. N = 8–12 per group. *p* < 0.05, influence of treatment (#) or genotype (*). VEH: vehicle.

**Figure 9 ijms-22-08907-f009:**
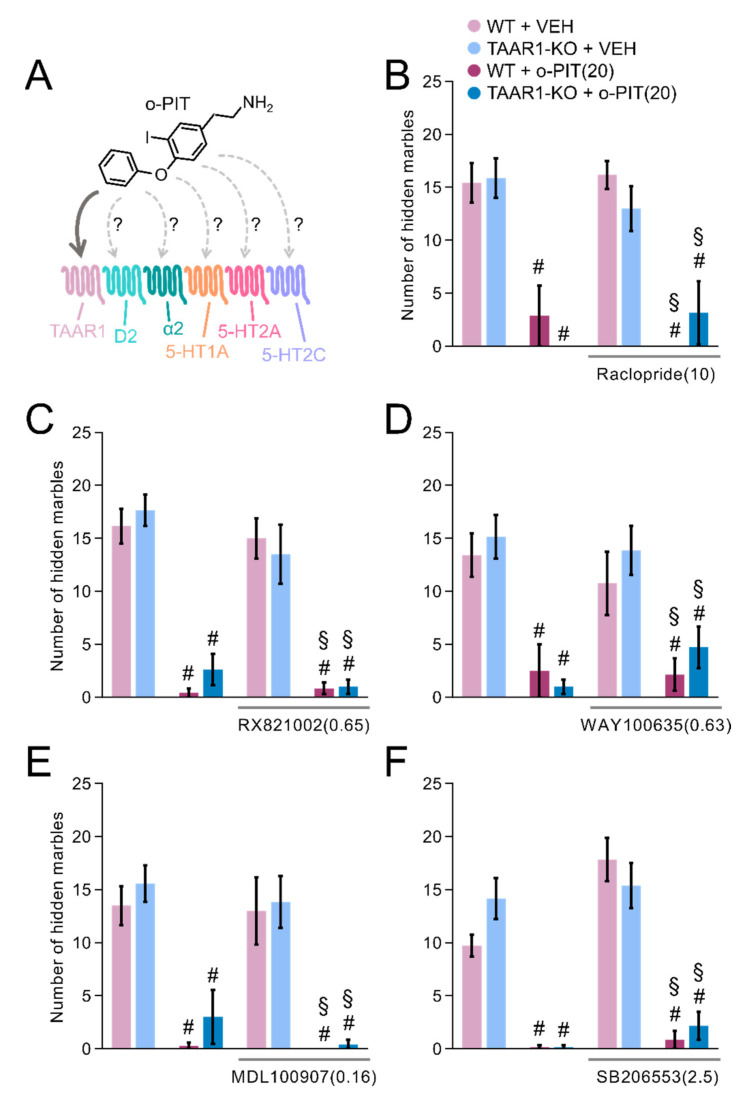
Influence of different monoaminergic receptors’ antagonists on o-PIT’s anticompulsive effects in TAAR1-KO and WT mice. (**A**) Schematic depiction of the possible monoaminergic receptors that may bind o-PIT. (**B**–**F**) Values correspond to the number of hidden marbles over 30 min after pretreatment of raclopride (10 mg/kg) (B), RX821002 (0.65 mg/kg) (**C**), WAY100635 (0.63 mg/kg) (**D**), MDL100907 (0.16 mg/kg) (**E**) or SB206553 (2.5 mg/kg) (**F**) and acute o-PIT treatment (20 mg/kg). Data represent means ± SEM. N = 8–12 per group. *p* < 0.05, vehicle + vehicle vs. vehicle/antagonist + o-PIT (#), antagonist + vehicle vs. antagonist + o-PIT (§). VEH: vehicle.

**Table 1 ijms-22-08907-t001:** Resting levels and o-PIT-induced changes in biogenic amines, acetylcholine and amino acids in dialysates of mPFC in freely moving mice. Values are mean ± SEM concentration not corrected for probe recovery. N ≥ 8 per groups. * *p* < 0.05 WT vs. TAAR1-KO.

		Resting Levels		o-PIT (10 mg/kg i.p.) [AUC; % of Vehicle]	
	(Units)	WT	TAAR1-KO	WT	TAAR1-KO
DA	(pM)	238.87 ± 30.13	203.49 ± 23.84	160 ± 13	109 ± 6 *
NA	(pM)	445.49 ± 25.79	411.87 ± 22.47	115 ± 5	106 ± 4
5-HT	(pM)	188.29 ± 15.19	209.08 ± 22.40	109 ± 8	102 ± 4
Acetylcholine	(nM)	0.92 ± 0.10	1.13 ± 0.14	98 ± 7	107 ± 20
Alanine	(µM)	0.90 ± 0.12	1.18 ± 0.21	94 ± 3	84 ± 4
Arginine	(µM)	0.38 ± 0.07	0.48 ± 0.10	89 ± 2	91 ± 4
Aspartate	(µM)	70.83 ± 20.45	69.98 ± 20.50	103 ± 5	105 ± 6
GABA	(µM)	10.74 ± 1.82	13.24 ± 3.19	87 ± 8	82 ± 8
Glutamate	(µM)	2.05 ± 0.71	1.64 ± 0.83	106 ± 3	101 ± 3
Glutamine	(µM)	9.55 ± 2.26	13.61 ± 3.73	106 ± 5	101 ± 5
Glycine	(µM)	1.08 ± 0.10	1.53 ± 0.29	93 ± 2	90 ± 2
Phenethylamine	(µM)	0.40 ± 0.08	0.32 ± 0.09	108 ± 3	114 ± 6
L-serine	(µM)	1.31 ± 0.18	1.58 ± 0.25	96 ± 2	96 ± 5
D-serine	(µM)	0.45 ± 0.09	0.35 ± 0.06	95 ± 4	107 ± 4
Taurine	(µM)	6.92 ± 1.64	6.99 ± 3.20	116 ± 6	105 ± 4

## Data Availability

Data are available upon reasonable request.

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
