# Peer review of "Trace Amine-Associated Receptor 1 Contributes to Diverse Functional Actions of O-Phenyl-Iodotyramine in Mice but Not to the Effects of Monoamine-Based Antidepressants"

_ijms, 2021, doi:10.3390/ijms22168907_

Round 1

Reviewer 1 Report

Тhis interesting, important and carefully executed study brings important new information on the effect of TAAR1 agonist, o-phenyl-iodotyramine (o-PIT) and antidepressants in wild type and TAAR1-KO mice. The authors convincingly demonstrated that o-PIT induced hypothermia, pre-

pulse inhibition response by o-PIT, as well as its induction of striatal tyrosine hydroxylase phosphorylation and elevation of extracellular DA in the prefrontal cortex were reduced in TAAR1-KO compared to wild-type mice.  Furthermore, the O-PIT was active in both forced-swim and marble-burying tests, and its effects were blunted in TAAR1-KO mice. At the same time, the behavioral and neurochemical effects of classical antidepressants were not affected in TAAR1-KO mice. The authors concluded that o-PIT is a useful tool for exploring the hypothermic and antidepressant-related effects of TAAR1. At the same time, clinically-used antidepressants do not require TAAR1 for expression of their antidepressant properties.

The manuscript is well written and methods are described with all the necessary details. The observations reported here are novel and provocative.  This manuscript is broad in content and of substantial interest to the readership. Further strengths of the manuscript include good experimental design, the wealth of data, and careful follow-up of findings with additional analyses.  This reviewer has no major criticism of the manuscript. 

Minor issues:

1.Please Introduce abbreviations NA and 5-HT in the beginning of introduction (Line 35).  I would suggest also using NE instead of NA.

2.Line 45 – should be G protein-coupled

  1. Please define DA on Line 49.
  2. Please discuss Corell et al., 2021 PMID: 3412711 report on long-term safety and effectiveness of Ulotaront in schizophrenia.
  3. Please state if experimenters were blinded as regard to animals and treatments used.

Author Response

Point-to-Point responses:

Reviewer 1:

Тhis interesting, important and carefully executed study brings important new information on the effect of TAAR1 agonist, o-phenyl-iodotyramine (o-PIT) and antidepressants in wild type and TAAR1-KO mice. The authors convincingly demonstrated that o-PIT induced hypothermia, pre-pulse inhibition response by o-PIT, as well as its induction of striatal tyrosine hydroxylase phosphorylation and elevation of extracellular DA in the prefrontal cortex were reduced in TAAR1-KO compared to wild-type mice. Furthermore, the O-PIT was active in both forced-swim and marble-burying tests, and its effects were blunted in TAAR1-KO mice. At the same time, the behavioral and neurochemical effects of classical antidepressants were not affected in TAAR1-KO mice. The authors concluded that o-PIT is a useful tool for exploring the hypothermic and antidepressant-related effects of TAAR1. At the same time, clinically-used antidepressants do not require TAAR1 for expression of their antidepressant properties.

The manuscript is well written and methods are described with all the necessary details. The observations reported here are novel and provocative. This manuscript is broad in content and of substantial interest to the readership. Further strengths of the manuscript include good experimental design, the wealth of data, and careful follow-up of findings with additional analyses. This reviewer has no major criticism of the manuscript.

Minor issues:

1.Please Introduce abbreviations NA and 5-HT in the beginning of introduction (Line 35).  I would suggest also using NE instead of NA.

Response: This is now corrected.

2.Line 45 – should be G protein-coupled

Response: This is now corrected.

3.Please define DA on Line 49.

Response: This is now corrected.

4.Please discuss Corell et al., 2021 PMID: 3412711 report on long-term safety and effectiveness of Ulotaront in schizophrenia.

Response. Thank you for the comment. We added a sentence regarding that on Line 75-76.

5.Please state if experimenters were blinded as regard to animals and treatments used.

Response: This is now mentioned on Line 412.

Reviewer 2 Report

The authors have made an interesting article that has the potential to be accepted in the journal, although they must correct many aspects of it, and require a major revision of the current version, which I will now list.
The first is about the discussion. It is an original article, which does not have any reference to any table or figure in the discussion, so the contribution can be considered discrete, looking more like a bibliographic review than a research article. Furthermore, the authors' reference in the discussion (line 342) is from currently unpublished data, which perhaps should be included in this manuscript for acceptance in the International Journal of Molecular Sciences.
The second indicates again that the manuscript has to be rewritten, since line 190 appears a reference to table 1, and in my version there is no table in the manuscript. These formal aspects have to be more careful when submitting an article for publication.
The third important aspect refers to animal welfare. The authors indicate in line 367 and 368, a Community Directive from 1986, which has been repealed more than 10 years ago, since the current regulations are from September 22, 2010, when the EU adopted the Directive 2010/63 / EU . The authors should clarify this point, if the approval and the experiment is prior to 2010, and what is the reason why it is being sent for publication a decade later. On the contrary, if they have an approval code and a committee that authorized the experimentation (including the forced swim test), this should be indicated. It should be taken into account that there are currently other alternative tests to the forced swim test, currently very frowned upon in today's society Nature 571, 456-457 (2019) doi: https://doi.org/10.1038/d41586-019- 02133-2, and that it can be replaced by other equivalent tests, or the selection of this test instead of another equivalent, such as the rotarod test or other equivalent tests, should be very well justified.
Other important aspects would be to indicate at least in the text the first time that all the abbreviations used appear, NA, VEH, ... It must be borne in mind that not all readers of this manuscript are experts in Neuroscience like the authors, and it facilitates the understanding of these readers.
There are other aspects to improve, as in figure 1 D, where it should be indicated that the samples come from WT animals or those that correspond,
In point 4.2, the authors should indicate the drugs that dissolve with lactic acid, and its concentration, as well as their inclusion at the same concentration of lactic acid in the control group, which I suspect is called vehicle (VEH).
For all this, the article requires a major revision to be accepted in the journal.

Author Response

Point-to-Point responses:

Reviewer 2:

The authors have made an interesting article that has the potential to be accepted in the journal, although they must correct many aspects of it, and require a major revision of the current version, which I will now list.

  1. The first is about the discussion. It is an original article, which does not have any reference to any table or figure in the discussion, so the contribution can be considered discrete, looking more like a bibliographic review than a research article.

Response: Thank you for your comment. We cite all the figures and the table in the results part of the manuscript. It is somewhat difficult to find proper places to cite all of them again in the Discussion section.

Furthermore, the authors' reference in the discussion (line 342) is from currently unpublished data, which perhaps should be included in this manuscript for acceptance in the International Journal of Molecular Sciences.

Response: We apologize for that. The sentences concerning the unpublished observations have now been removed.

The second indicates again that the manuscript has to be rewritten, since line 190 appears a reference to table 1, and in my version there is no table in the manuscript. These formal aspects have to be more careful when submitting an article for publication.

Response: We apologize for this mistake. Table 1 is now added to the text.

The third important aspect refers to animal welfare. The authors indicate in line 367 and 368, a Community Directive from 1986, which has been repealed more than 10 years ago, since the current regulations are from September 22, 2010, when the EU adopted the Directive 2010/63 / EU . The authors should clarify this point, if the approval and the experiment is prior to 2010, and what is the reason why it is being sent for publication a decade later. On the contrary, if they have an approval code and a committee that authorized the experimentation (including the forced swim test), this should be indicated. It should be taken into account that there are currently other alternative tests to the forced swim test, currently very frowned upon in today's society Nature 571, 456-457 (2019) doi: https://doi.org/10.1038/d41586-019- 02133-2, and that it can be replaced by other equivalent tests, or the selection of this test instead of another equivalent, such as the rotarod test or other equivalent tests, should be very well justified.

Response: We have now revised and added text related to ethical permissions on lines 414-417: “All experiments were conducted in accordance with European/EU (Directive 2010/63) directives on animal experimentation and with local (internal/Servier Research Institute or Stockholm Animal Ethics Committee 18002-2017) ethical committee approvals.”

We have added a sentence on lines 390-391 that additional behavioral tests relevant for antidepressant actions would be interesting to perform in the future.

Other important aspects would be to indicate at least in the text the first time that all the abbreviations used appear, NA, VEH, ... It must be borne in mind that not all readers of this manuscript are experts in Neuroscience like the authors, and it facilitates the understanding of these readers.

Response: This is now corrected

There are other aspects to improve, as in figure 1 D, where it should be indicated that the samples come from WT animals or those that correspond,

Response: The figure 1D shows the histoenzymological detection of lacZ insert in TAAR1-KO. We have now indicated in the figure legend that the pictures are taken from TAAR1-KO brain sections and that the staining represents expression under the endogenous TAAR1 promoter. 

In point 4.2, the authors should indicate the drugs that dissolve with lactic acid, and its concentration, as well as their inclusion at the same concentration of lactic acid in the control group, which I suspect is called vehicle (VEH).

Response: This is now indicated in the materials and methods section (lines 422-424).

Round 2

Reviewer 2 Report

Congratulations to the authors for this new version of the manuscript, which can be accepted in its current version.

There are techniques used with animals that are questionable, although if admitted by the ethics committee in 2017, for me it raises no more doubts about the need to perform, instead of being replaced by other equivalents.

The inclusion of the table and modifications presented in this new version are important and improve the quality of the work, in addition to updating it.

On the other hand, the editorial team considers that it fits the scope of the journal, and they provide important information that can advance the treatment of many patients with depression.

For all this, the article can be accepted in its current version.